# Inhibitors of NAD^+^ Production in Cancer Treatment: State of the Art and Perspectives

**DOI:** 10.3390/ijms25042092

**Published:** 2024-02-08

**Authors:** Moustafa S. Ghanem, Irene Caffa, Fiammetta Monacelli, Alessio Nencioni

**Affiliations:** 1Department of Internal Medicine and Medical Specialties (DIMI), University of Genoa, Viale Benedetto XV 6, 16132 Genoa, Italy; irene.caffa@unige.it (I.C.); fiammetta.monacelli@unige.it (F.M.); 2Ospedale Policlinico San Martino IRCCS, Largo Rosanna Benzi 10, 16132 Genova, Italy

**Keywords:** NAD^+^, cancer metabolism, Preiss-Handler pathway, NAPRT, NMNAT, NADSYN, NAMPT, inhibitors

## Abstract

The addiction of tumors to elevated nicotinamide adenine dinucleotide (NAD^+^) levels is a hallmark of cancer metabolism. Obstructing NAD^+^ biosynthesis in tumors is a new and promising antineoplastic strategy. Inhibitors developed against nicotinamide phosphoribosyltransferase (NAMPT), the main enzyme in NAD^+^ production from nicotinamide, elicited robust anticancer activity in preclinical models but not in patients, implying that other NAD^+^-biosynthetic pathways are also active in tumors and provide sufficient NAD^+^ amounts despite NAMPT obstruction. Recent studies show that NAD^+^ biosynthesis through the so-called “Preiss-Handler (PH) pathway”, which utilizes nicotinate as a precursor, actively operates in many tumors and accounts for tumor resistance to NAMPT inhibitors. The PH pathway consists of three sequential enzymatic steps that are catalyzed by nicotinate phosphoribosyltransferase (NAPRT), nicotinamide mononucleotide adenylyltransferases (NMNATs), and NAD^+^ synthetase (NADSYN1). Here, we focus on these enzymes as emerging targets in cancer drug discovery, summarizing their reported inhibitors and describing their current or potential exploitation as anticancer agents. Finally, we also focus on additional NAD^+^-producing enzymes acting in alternative NAD^+^-producing routes that could also be relevant in tumors and thus become viable targets for drug discovery.

## 1. Introduction

### 1.1. NAD^+^ and Cancer Metabolism

Nicotinamide adenine dinucleotide (NAD^+^) is a ubiquitous metabolite that performs indispensable roles to maintain cellular homeostasis. Being involved in more than 500 cellular reactions, NAD^+^ acts either as a redox cofactor or as an enzymatic substrate, coordinating a vast array of vital processes inside the cell [1,2,3,4,5]. The most renowned cellular function of NAD^+^ is that it serves as a redox cofactor for numerous dehydrogenases that participate in crucial metabolic and bio-energetic cellular events such as glycolysis, the tricarboxylic acid (TCA) cycle, oxidative phosphorylation (OXPHOS), and fatty acid oxidation [1,2,3,4,5]. NAD^+^ is also utilized as a substrate by a heterogeneous group of NAD^+^-consuming enzymes represented by the poly-(ADP Ribose) polymerases (PARPs), the NAD^+^-dependent deacetylases sirtuins (SIRT1-7), and the NAD^+^-dependent glycohydrolases and ADP-ribosyl cyclases CD38 and CD157, and sterile alpha and toll/interleukin-1 receptor motif containing 1 (SARM1) [1,2,3,4,5]. These enzymes regulate fundamental cellular processes including DNA damage repair, gene expression, cell signaling, calcium mobilization, apoptosis, circadian rhythm, and inflammatory responses [1,2,3,4]. As a cofactor, NAD^+^ acts as an electron carrier that oscillates back and forth between its oxidized form and its reduced form (i.e., NADH), but its molecular backbone remains intact. In contrast, as a substrate, NAD^+^ is degraded by the aforementioned NAD^+^-consuming enzymes, always yielding nicotinamide (NAM) as a byproduct.

Neoplasms are almost invariably characterized by heightened NAD^+^ demands primarily to sustain their altered metabolic requirements [6,7,8,9]. In this context, reprogrammed metabolism has been recognized as one of the hallmarks of oncogenic transformation [10]. The changes in cell metabolism that characterize many tumors are fundamental in order for them to support their continuous growth and proliferative demands [11,12]. Glucose metabolism is almost inevitably skewed in tumors [13]. Normal cells typically utilize glucose to fuel energy production: once glucose is imported into the cell, it is primarily metabolized in the cytosol via a process called glycolysis, in which a cascade of 10 enzymatic reactions mediate the breakdown of one glucose molecule into two pyruvate molecules with a net generation of two NADH molecules and two adenosine triphosphate (ATP) molecules. In the presence of oxygen, pyruvate enters the mitochondria and is transformed into acetyl-CoA. Here, the so-called cellular respiration takes place with the occurrence of the TCA cycle and OXPHOS ultimately resulting in the generation of a large amount of energy from the metabolism of glucose-derived acetyl-CoA (around 34 ATP molecules). On the other hand, under anaerobic conditions, fermentation takes place rather than mitochondrial respiration where pyruvate is reduced into lactate in the cytosol by lactate dehydrogenase.

In comparison to normal cells, glucose uptake in tumor cells is enhanced as a consequence of the activity of oncogenes (e.g., those from the PI3K/AKT pathway) that upregulate the cell surface expression of glucose transporter GLUT1 [14], or of the loss of tumor suppressors, such as SIRT6, which leads to increased GLUT1 and GLUT4 expressions and to a reduced carbon flux into mitochondrial respiration (through an increased pyruvate dehydrogenase kinase expression) [15,16,17,18]. In addition, in cancer cells, glycolysis is highly active even when oxygen is available. This phenomenon was described by Otto Warburg in the early decades of the 1900s and is known as aerobic glycolysis (or the Warburg effect) [19]. Since aerobic glycolysis is much less efficient than cellular respiration in terms of energy production (2 and 34 ATP molecules are produced through glycolysis and through mitochondrial respiration, respectively), the reason why cancer cells would rely so much on this metabolic process was not of immediate understanding. It was initially postulated that tumors depend on glycolysis as a result of defective mitochondria. However, this hypothesis was largely refuted by emerging evidence demonstrating that mitochondrial machinery is not only intact in several tumors, but also is instrumental in bolstering tumorigenesis [20,21,22]. It was only in the late 2000s that it became clear that glycolysis is exploited by cancer cells to obtain intermediates that can act as building blocks to support their unchecked growth and proliferation. Considerable amounts of glycolytic intermediates are funneled toward alternative metabolic pathways (in particular the pentose phosphate pathway), which are essential to building up the tumor biomass. Since both an elevated glycolytic flux and the mitochondrial metabolism require NAD^+^, cancer cells need to maintain high NAD^+^ levels compared to their non-transformed counterparts. In addition, it is important to note that tumor cell proliferation and survival can also be critically influenced by the NAD^+^-consuming enzymes and the processes that they coordinate (for example, PARP-mediated DNA damage repair and the sirtuin-dependent transcriptional and post-transcriptional regulations of P53 and other targets). Moreover, NAD^+^ phosphorylation by NAD^+^ kinase (NADK) produces NADP, which, together with its reduced form NADPH, acts as a key antioxidant [23,24]. Cancer cells also have increased demands for NADP(H) (and thus for NAD^+^) to combat the excessive accumulation of reactive oxygen species (ROS), but also to drive mitochondrial glutamine metabolism through the TCA cycle (a process called anaplerosis), which in turn, provides building blocks for other tumor anabolic processes such as lipid biosynthesis [23,24].

Cancer cells are particularly sensitive to NAD^+^ depletion, which strongly affects cell metabolism and energy production, and also other critical processes, such as DNA damage repair, immune escape, the ability to scavenge ROS, pro-oncogenic signaling pathways, and oncogene expression. Ultimately, these effects result in cancer cell demise. Therefore, exploiting these liabilities of cancer cells through approaches that deplete NAD^+^ is now considered an appealing therapeutic strategy. The most widely studied strategy to deplete intracellular NAD^+^ in cancer cells is to interfere with their NAD^+^ biosynthetic machinery.

### 1.2. Mammalian NAD^+^ Biosynthesis

Mammalian cells can meet their NAD^+^ requirements through multiple NAD^+^-generating metabolic pathways that start from either vitamin B3 or from the essential amino acid tryptophan (Figure 1). Vitamin B3 comprises three different forms, which are NAM, nicotinic acid (NA), and nicotinamide riboside (NR) [25]. Each of these forms serves as a starting building block for NAD^+^ formation through a distinct metabolic route: NAM and NA generate NAD^+^ via the salvage pathway and the Preiss-Handler (PH) pathway, respectively, while their nucleoside forms (i.e., NR and nicotinic acid riboside (NAR)) generate NAD^+^ via an alternative nucleoside pathway. Tryptophan generates NAD^+^ via the de novo pathway (also known as the kynurenine pathway), which is active mainly in hepatic and renal tissues [26]. Although NAD^+^ biosynthesis in mammalian cells comprises an intricate network of precursors and pathways, it generally follows a scheme that consists of two main steps: (1) the generation of a mononucleotide from a simple NAD^+^ precursor or building block. In the NAM salvage pathway, NAM is converted into nicotinamide mononucleotide (NMN) through the catalytic activity of nicotinamide phosphoribosyltransferase (NAMPT). In the PH pathway, a parallel enzyme called nicotinate phosphoribosyltransferase (NAPRT) catalyzes the conversion of NA into nicotinic acid mononucleotide (NAMN) [27,28]. A third phosphoribosyltransferase enzyme named quinolinate phosphoribosyltransferase (QPRT) also produces NAMN from quinolinic acid (QA), which is an intermediate that is obtained from tryptophan metabolism in the de novo pathway. It is worth noting that each of these phosphoribosyltransferase enzymes is the rate-limiting enzyme of its own NAD^+^-generating route. In addition, the enzyme nicotinamide riboside kinase (NMRK) is key for the cell to be able to utilize NR and NAR as NAD^+^ precursors: the former is converted by NMRK into NMN, while the latter, thanks to NMRK activity, becomes NAMN [29,30]. Interestingly, NMN could be cleaved in the extracellular milieu by the ecto-enzyme CD73 to produce NR, which in turn, can enter the cells and eventually support intracellular NAD^+^ formation [31,32]. Thus, this metabolic conversion that takes place outside the cells represents an important mechanism for sustaining NAD^+^ biosynthesis [31,32]. It is worth noting that the reduced forms of NR and NMN were recently identified as novel NAD^+^ precursors that can elevate cellular and systemic NAD^+^ levels [33,34]. Adenosine kinase rather than NMRK mediates the phosphorylation of reduced NR into reduced NMN [35]. (2) The second step consists of the conversion of the mononucleotide moieties into their respective dinucleotides: all of the NAD^+^ metabolic routes intersect at the level of this enzymatic step, which is catalyzed by the enzymes nicotinamide/nicotinate mononucleotide adenylyltransferases 1–3 (NMNAT1-3). NMNATs transform NMN into NAD^+^ and NAMN into nicotinic acid adenine dinucleotide (NAAD) using ATP as the source of the adenylyl moiety. In the de novo and in the PH pathways, as well as when NAR is used as a starting block for NAD^+^ production, an additional step is necessary for NAD^+^ formation, i.e., the replacement of the carboxylic group of NAAD with an amide group. This reaction is catalyzed by the enzyme NAD^+^ synthetase (NADSYN1) and the amino acid glutamine serves as the donor of the amide nitrogen in it. NAM is produced by all NAD^+^-consuming enzymes as a byproduct and is then either recycled to rebuild NAD^+^ or modified by nicotinamide N-methyltransferase (NNMT) to yield 1-methylnicotinamide (MNAM) (particularly in the liver), which is ultimately excreted in urine [36,37]. NAM reconversion into NAD^+^ is a ubiquitous process in mammalian tissues. Accordingly, NAMPT was found to be expressed in all types of tissues that were investigated, and the NAM salvage pathway is conceived to be the predominant pathway for NAD^+^ production. In this review, we will describe the relevance of these NAD^+^-producing enzymes as potential targets in cancer, mainly focusing on the reported molecules that inhibit the activity of NAD^+^-producing enzymes other than NAMPT (which has been extensively covered in other recent articles). In addition, the potential challenges that might be encountered in the employment of these inhibitors as drug candidates in oncology will also be highlighted.

## 2. Targeting Nicotinamide Phosphoribosyltransferase

Cancer cells primarily depend on the salvage pathway to produce the NAD^+^ they need for their proliferation and survival, and thus, they are expected to be vulnerable to NAMPT inhibition. This notion was substantiated by several studies that reported the overexpression of NAMPT in a myriad of hematological and solid malignancies [8,9]. Numerous reports also highlighted an association between high NAMPT levels and poor clinical outcomes, including worse survival, for different types of cancer [38,39,40]. Accordingly, NAMPT is the NAD^+^-producing enzyme against which the largest number of inhibitors were developed over the past two decades (Figure 1). NAMPT inhibitors displayed remarkable anticancer activity in cellular and animal models of cancer [8]. Amongst the identified NAMPT inhibitors, the prototypical compound FK866 (also known as (E)-Daporinad) [41], CHS-828 [42,43], and its prodrug GMX1777 [44] were evaluated in cancer patients in early-phase clinical trials. In these clinical studies, NAMPT inhibitors showed marginal or no tumor responses as well as side effects such as thrombocytopenia and lymphopenia (in the case of FK866) or gastrointestinal side effects (in the case of the orally administered CHS-828) [45,46,47,48]. The failure of the first NAMPT inhibitors in the clinic sparked efforts to optimize the use of these agents, which culminated in a second wave of NAMPT inhibitors (represented by OT-82 [49] and the dual NAMPT-PAK4 inhibitor KPT9274 [50]) being recently assessed in clinical trials. Several review articles discuss the medicinal chemistry aspects of the development of NAMPT inhibitors and also the emerging roles of NAMPT beyond being an enzyme that generates NAD^+^ (since NAMPT also exists as an extracellular form mainly acting as a cytokine that was found to mediate multiple pro-oncogenic roles) [51,52,53,54,55]. We have also recently reviewed the preclinical and clinical aspects of NAD^+^-lowering drugs, with a special focus on NAMPT inhibitors, including their downstream effects and their optimization strategies [8]. Here, we will just remind the reader that the last two years have witnessed the emergence of new small-molecule chemical NAMPT inhibitors that exhibited very potent anticancer activity against hematological malignancies with IC50 values in the picomolar range [56,57,58]. A synergistic interaction between FK866 and the antidiabetic drug metformin was also recently reported in pancreatic cancer cells [59]. In addition to the conventional small-molecule inhibitors that block NAMPT enzymatic activity, one of the most promising approaches that have recently emerged is to degrade NAMPT by triggering its ubiquitin-mediated proteolysis. Compounds developed with this novel technology are named proteolysis-targeting chimeras (PROTACs) [60]. They typically consist of two protein-binding domains (one domain binds to E3 ubiquitin ligase and the second domain binds to the target protein, which is NAMPT in this case) connected by a linker. PROTACs B3 and B4 showed efficient NAMPT degradation and marked in vitro and in vivo anticancer activity against A2780 ovarian cancer cells [61,62]. Two other NAMPT-degrading PROTACs showed superior antileukemia activity compared to FK866 by targeting both intracellular and extracellular forms of NAMPT [63]. Similarly, the NAMPT-degrading PROTAC A7 depleted intracellular NAMPT, reduced the levels of secreted NAMPT, and elicited remarkable antitumor responses in mouse tumor models [64]. The antitumor effects of PROTAC A7 were mostly achieved through antitumor immunity activation, which in turn, resulted from dampened activity of tumor-infiltrating myeloid-derived suppressor cells [64]. Taken together, these results highlight an advantage of the NAMPT-degrading PROTAC technology compared to the conventional inhibitors, which is their ability to eliminate both intracellular and secreted NAMPT, thus blocking not only the enzymatic function of NAMPT but also its extracellular pro-tumorigenic roles (that are independent of its enzymatic activity [65]). Finally, novel antibody–drug conjugates with NAMPT inhibitors as payloads have suppressed tumor growth in xenograft models of breast cancer (HER2-expressing MDA-MB-453 cells) and leukemia (B7H3-expressing THP-1 cells) [66].

## 3. Targeting Nicotinate Phosphoribosyltransferase

### 3.1. NAPRT as a Target in Cancer

Resistance to NAMPT inhibitors is a significant clinical challenge. Some cancer types can use alternative NAD^+^-generating pathways and as a consequence maintain sufficient NAD^+^ levels when the primary NAM salvage pathway is obstructed. In these tumors, the cytotoxic activity of NAMPT inhibitors is significantly compromised. Among these alternative NAD^+^-producing routes, the PH route appears to be frequently exploited by tumors. Accordingly, the rate-limiting enzyme of this NAD^+^ production pathway, NAPRT, has emerged as a promising target in cancer treatment. NAPRT drives the first enzymatic step in the PH pathway, by catalyzing the conjugation of the phosphoribosyl group from phosphoribosyl pyrophosphate (PRPP) to NA, thereby yielding NAMN. In turn, NAMN is subsequently transformed into NAAD and then, finally, into NAD^+^. Hara and co-workers demonstrated that NAPRT expression mediated the effects of NA supplementation on human cells in terms of raising intracellular NAD^+^ contents and in terms of protecting against oxidative stress [67]. The same authors were also able to detect high NAPRT levels in murine tissues, including the small intestine, liver, and kidney [67]. An extensive study that later explored NAMPT and NAPRT expression patterns across normal human tissues and cancer cell lines found that both *NAMPT* and *NAPRT* transcripts were widely expressed across normal human tissues [68]. While in tumors, NAMPT is ubiquitously expressed, NAPRT expression levels were found to be largely variable with several cancer cell lines showing either marginal or no NAPRT expression and many other cell lines exhibiting high NAPRT levels [68]. To investigate how cancer cells select their NAD^+^ biosynthetic route, Chowdhry and colleagues conducted a comprehensive analysis of more than 7000 tumors and 2600 matched normal samples, spanning 19 tissue types [69]. These authors concluded that tumors that stem from normal tissues with elevated NAPRT expressions usually show amplifications in the *NAPRT* gene and hence become reliant on the PH pathway for their NAD^+^ metabolism [69]. Moreover, tumors that are addicted to the NAM salvage pathway originate from tissues with negligible NAPRT expression [69]. The mechanisms regulating the NAPRT expression in tumors have been addressed by several studies. In the first place, as anticipated above, *NAPRT* gene amplification is well documented in a wide spectrum of solid tumors, including, but not limited to, ovarian, prostate, and pancreatic cancer (all showing an amplification frequency between 25% and 35% of the studied cases), and this correlates with a high expression of *NAPRT* mRNA [69,70]. In other tumors, the NAPRT expression is epigenetically modulated, frequently, but not always, to dampen it. One of the well-established epigenetic mechanisms by which tumors lose NAPRT expression is through the hypermethylation of the *NAPRT* promoter’s CpG islands. This mechanism was first described by Shames and colleagues in a large panel of cell lines of non-small cell lung cancer [71]. Importantly, this finding unveiled important therapeutic implications where a synthetic lethal interaction was observed between NAPRT deficiency and NAMPT inhibition. Consistently, the methylation of the *NAPRT* promoter was detected in several chondrosarcoma cell lines and was correlated with a reduced NAPRT expression that, in turn, was associated with enhanced sensitivity to NAMPT inhibitors [72]. Further evidence was provided by Lee and colleagues who noted that in numerous gastric cancer cell lines, the presence of markers of the epithelial-to-mesenchymal transition (EMT) was associated with a diminished NAPRT expression (again, due to *NAPRT*-promoter hypermethylation) [73]. As a result, these NAPRT-deficient cancers are also particularly vulnerable to FK866 [73]. These authors postulated that NAPRT loss in these cell lines would drive EMT by activating the Wnt/β-catenin signal [73]. Likewise, an inverse relation between the expressions of EMT markers and NAPRT were also seen in pancreatic and colorectal cancer cells [74]. Interestingly, *NAPRT*-promoter hypermethylation in cancers could result from other specific mutations. For instance, isocitrate dehydrogenase 1 (IDH1)-mutated gliomas typically show diminished NAPRT expression due to *NAPRT* promoter hypermethylation [75]. A similar finding was recently discovered in pediatric glioma models that harbor mutations in the *protein phosphatase Mg2^+^/Mn2^+^ dependent 1D* (*PPM1D*) gene [76]. Again, extreme sensitivity to the NAMPT inhibitors was illustrated in these two mutated subtypes of brain tumors [75,76]. The modulation of histone methylation is another epigenetic mechanism that can affect NAPRT expression in uveal melanoma [77]. In this ocular malignancy, NAPRT expression was found to be enhanced, instead of reduced, by the methyltransferase, disruptor of telomeric silencing-1-like (DOT1L), i.e., via enhanced H3 methylation at lysine 79 (H3K79) [77]. In turn, an increased NAPRT expression was shown to foster malignant transformation by fueling NAD^+^ biosynthesis [77]. A recent study illustrated that the chromatin modifier and epigenetic regulator bromodomain-containing protein 4 (BRD4) plays an important role in controlling the NAPRT expression in hepatocellular carcinoma (HCC) cells since NAPRT levels were reduced in HCC cell lines upon treatment with the BRD4 inhibitor AZD5153 [78].

A NAPRT expression or lack of NAPRT expression in tumors has two types of implications. In the first place, studies have shown that NAPRT-proficient tumors are highly reliant on NAPRT-mediated NAD^+^ biosynthesis, which has pro-oncogenic effects in terms of promoting protein synthesis, fostering energy production and mitochondrial OXPHOS, supporting DNA repair (especially in BRCA-deficient cancer cells), and modulating tumor cell responsiveness to DNA-damaging agents and NAMPT inhibitors [70]. In agreement with these insights, the growth of PH-amplified OV4 ovarian cancer xenografts was completely repressed upon NAPRT depletion [69]. We also demonstrated that *NAPRT* silencing sensitized NAPRT-proficient ovarian and pancreatic cancer cell lines to FK866 treatment, whereas the NAPRT-proficient wild-type counterparts were completely refractory to NAMPT inhibition [70]. Therefore, in NAPRT-proficient tumors, the goal will be to inhibit this enzyme in order to make NAMPT inhibitors and possibly other types of anticancer drugs more active. On the other hand, this lack of NAPRT expression is a liability of cancer cells that can also be therapeutically exploited: tumors that lack NAPRT should not be able to utilize NA to replenish their NAD^+^ content. In these instances, administering NAMPT inhibitors in combination with NA is postulated to help many healthy, NAPRT-proficient tissues, but not NAPRT-deficient cancer cells in avoiding the consequences of NAMPT obstruction. This treatment approach was indeed successful in several NAPRT-deficient tumor models, expanding the therapeutic window of NAMPT inhibitors and mitigating toxicity [43,71,79,80]. A lack of NAPRT expression in tumors serves as a biomarker that dictates which tumors can benefit from this approach [71,79]. It should be mentioned that in one study utilizing NA in combination with the NAMPT inhibitor GNE-617, the antitumor activity of the latter was found to be prevented by the concomitant NA administration in NAPRT-deficient tumor xenograft models, which was ascribed to an increased NAM and NAD^+^ production in the liver with the consequent rise in tumor NAM and NAD^+^ levels. Elevated NAM levels inside the tumors might compromise the antiproliferative activity of competitive NAMPT inhibitors by partially reactivating the NAMPT-dependent NAD^+^-generating pathway [81]. Thus, further studies are required to confirm the usefulness of a combined NAMPT inhibitor plus NA as a treatment for NAPRT-deficient neoplasms. Obviously, the use of a combined NAMPT inhibitor plus NA supplementation is not foreseen in the case of PH-amplified tumors since in these instances, NA supplementation will abrogate the effect of NAMPT inhibitors (rather, in these cases, NAMPT inhibitors should probably be coupled with NAPRT inhibitors or NA deprivation). It is also worth noting that a recent study found that NA supplementation ameliorates cancer cachexia, a condition that is commonly diagnosed in patients with cancer and that is frequently worsened through chemotherapy [82]. This effect of NA was ascribed to its ability to boost NAD^+^ in skeletal muscles and to enhance mitochondrial metabolism, although whether this effect is mediated directly by activating the PH pathway in skeletal muscles is not completely clear [82].

### 3.2. NAPRT Inhibitors

Based on the above insights, there is a justified need for developing NAPRT inhibitors to be utilized together with NAMPT inhibitors against PH-activated tumors. While the development of NAMPT inhibitors is proceeding at a fast pace, the identification of drugs that target NAPRT (and also other NAD^+^-biosynthetic enzymes) is still in the nascent stage (Figure 2). Surprisingly, however, inhibitors of the NAPRT enzyme (summarized in Table 1) were first described much earlier than NAMPT inhibitors. Nearly fifty years ago, Gaut and Solomon discovered that the accumulation of radioactivity inside the human blood platelets that were incubated with radiolabeled isotopic NA was suppressed by several NA analogs [among which 2-hydroxynicotinic acid (2-HNA) was the most potent] as well as metabolic inhibitors like dinitrophenol, NaF, NaCN, and salicylic acid [83]. In particular, 2-HNA inhibited the incorporation of radiolabeled NA into NAD^+^, NAM, and other unidentified compounds that were presumed to be intermediates of NAD^+^ biosynthesis, and this effect was concentration-dependent [83]. Shortly after, the same authors conducted kinetic studies that characterized the incorporation of radiolabeled NA into NAMN by human platelets lysates with or without the addition of several NA analogs and found 2-HNA to act as a NAPRT inhibitor with an apparent inhibition constant (Ki) of 230 µM [84]. Using the same experimental methodology, additional NAPRT inhibitors were annotated, including pyrazinoic acid (with an apparent Ki of 75 µm), 2-fluoronicotinic acid (apparent Ki = 280 µm), and several non-steroidal anti-inflammatory drugs (NSAIDs) such as flufenamic acid (Ki = 46 µM), mefenamic acid (Ki = 76 µM), and salicylic acid and phenylbutazone (both had an apparent Ki = 160 µM) [84,85]. The identification of NAPRT inhibitors afterward almost halted for nearly four decades until Galassi and colleagues, using a human recombinant NAPRT enzyme and an HPLC method to detect NAMN formation, reported a series of metabolites that inhibited NAPRT enzymatic activity at millimolar concentrations [86]. CoA was the most effective NAPRT-inhibiting metabolite (with an IC50 value of around 850 µM) followed by several acyl-CoA derivatives, namely succinyl-CoA, glutaryl-CoA, and acetyl-CoA [86]. Less pronounced NAPRT inhibitory effects were also reported with three glycolysis intermediates: fructose 1,6-bisphosphate, phosphoenolpyruvate, and glyceraldehyde 3-phosphate (all showing comparable percentages of enzyme inhibition at 1 mM), whereas dihydroxyacetone phosphate (DHAP) and pyruvate, the end product of glycolysis, showed a stimulatory effect on NAPRT enzymatic activity [86]. Interestingly, ATP displayed mixed stimulatory and inhibitory effects on NAPRT activity depending on the substrates’ saturation levels [86]. In the same study, the authors provided a predicted model for the 3D structure of the human NAPRT enzyme, but the actual 3D crystal structure of human NAPRT was resolved and revealed only three years later by Marletta et al. [87]. Until that time, none of these NAPRT-inhibiting molecules had been tested as agents against tumors. The first evidence of an anticancer activity attributed to an NAPRT inhibitor was reported by our group in 2017 when we demonstrated that the prototypical NAPRT inhibitor 2-HNA cooperates with FK866 in blunting NAD^+^ levels in the NAPRT-proficient ovarian cancer OVCAR-5 cells. Co-treatment with 2-HNA and FK866 prompted marked cell death of OVCAR-5 (ovarian cancer) and of Capan-1 (pancreatic cancer) cells, which both express NAPRT in abundant amounts (thereby recreating the effects of *NAPRT* silencing) [70]. In OVCAR-5 xenograft-bearing mice, combining FK866 with the sodium salt form of 2-HNA resulted in a significant prolongation of mice survival that could not be attained in mice treated with FK866 only [70]. Similarly, it has been recently shown that the antitumor effects of the NAMPT inhibitors GNE-617 and GMX1778 were enhanced when coupled to 2-HNA in two in vivo models of head and neck squamous cell carcinomas [88]. Given the emerging relevance of NAPRT inhibitors in cancer therapy, considerable efforts have been devoted to discovering more molecules that inhibit NAPRT (summarized in Table 1). We recently described two additional NAPRT inhibitors through a high-throughput in silico screening of a chemical library composed of more than 500,000 compounds, taking advantage of the available 3D crystal structure of human NAPRT [87,89]. The two compounds demonstrated NAPRT inhibitory activity in enzymatic assays [89]. Further biochemical experiments on the purified recombinant NAPRT enzyme illustrated that compound **8** (4-hydroxynicotinic acid) acts as a competitive NAPRT inhibitor, whereas compound **19** is a non-competitive NAPRT inhibitor (the Ki values of compounds **8** and **19** were 307.5 and 295 µM, respectively) [89]. In cellular models, the two compounds demonstrated anticancer activity at 0.1 mM in cell growth inhibition assays, where the sensitivity of FK866 to NAPRT-expressing ovarian cancer (OVCAR-5 and OVCAR-8) and colorectal cancer (HCT116) cell lines could be restored [89]. Among the two inhibitors, compound **8** was more effective and demonstrated an ability to sensitize cancer cells to FK866 that was comparable to that of 2-HNA [89]. In addition, compound **8** showed favorable pharmacokinetic parameters and drug-like properties as predicted using in silico tools [89]. A similar approach was adopted by Franco and colleagues who started from a structure-based computational approach for NAPRT inhibitor identification and found IM 29 (a compound with a 1,3-benzodioxole structural backbone) as a lead compound from their screens [90]. In enzymatic and cell-based assays, IM 29 was found to inhibit human recombinant NAPRT enzyme activity (IC50 of 160 µM) and to cooperate with FK866 in reducing NAD^+^ levels in OVCAR-5 cells [90].

A continuous fluorometric enzymatic assay was recently devised and validated, thereby providing a useful tool to easily and rapidly screen putative NAPRT inhibitors [91]. This assay is based on a fluorometric method that can, in a single step, detect the formation of NADH starting from the NAPRT-catalyzed reaction product NAMN [91]. In the validation experiments of this assay, the NAPRT inhibitory activities of 2-HNA, 2-fluoronicotinic acid, pyrazinoic acid, and salicylic acid (which were previously described as NAPRT inhibitors by Gaut and Solomon) were confirmed [91]. According to this assay, the Ki values of these four NAPRT inhibitors varied between 149 and 215 µM. Interestingly, besides 2-HNA, pyrazinoic acid was also found to sensitize OVCAR-5 cells to FK866 by reducing viability and blunting NAD^+^ levels inside the cells [91]. In the same cellular model, no significant effect on cell viability was noted when coupling 2-fluoronicotinic acid with FK866, while salicylic acid treatment was suggested to elicit an unspecific cytotoxic activity [91]. Shortly after, the same group employed the assay to screen more than 200 small molecules for their activity on the recombinant NAPRT enzyme and succeeded in identifying a screening hit, compound **17**, that bears a 1,2-dimethylbenzimidazole moiety [92]. Subsequent medicinal chemistry efforts and structure–activity relationship studies starting from this hit compound led to the design of several structural analogs, among which compound **18** was demonstrated to inhibit NAPRT more effectively than its parent, compound **17** (46% and 30% NAPRT inhibition with compound **18** and compound **17**, respectively, when both were used at a 1 mM concentration) [92]. Mechanistically, compound **18** was shown to act via non-competitive inhibition toward NA (Ki = 338 µM) and through mixed inhibition toward PRPP (Ki = 134 µM) [92]. In silico docking studies that interrogated the binding mode of compound **18** were in agreement with the described mode of action [92]. From the perspective of drug development, compound **18** exhibited favorable pharmacokinetic properties including high kinetic solubility at physiological pH, low protein binding, good metabolic stability, and cell permeability [92]. The only reported limitation is that compound **18** is a P-gp substrate, which may limit its accumulation inside the tumors [92]. Interestingly, the same study also revealed the first series of NAPRT activators [92]. Given that NAPRT activity could not suppressed by NAD^+^ (unlike NAMPT) [67], NAPRT activators could have potential clinical applications, particularly in conditions where boosting NAD^+^ is of therapeutic relevance. Despite these recent efforts, the concentrations at which all the available NAPRT inhibitors demonstrate activity remain in the 3-digit micromolar-to-millimolar range, and no NAPRT inhibitor has progressed to clinical studies so far. Therefore, there remains a critical need to produce more potent NAPRT inhibitors with drug-like features.

## 4. Targeting Nicotinamide/Nicotinate Mononucleotide Adenylyltransferase

### 4.1. NMNAT as a Target in Cancer

The second step in the PH pathway of NAD^+^ generation is governed by the NMNAT enzyme, which in mammalian cells, exists in three isomeric forms (i.e., NMNAT1, 2, and 3) [93]. NMNAT mediates the transfer of an adenylyl group from ATP to the mononucleotides NMN and NAMN, thus converting them into their corresponding dinucleotides, NAD^+^ and NAAD, respectively. Being the common enzyme that is shared by all NAD^+^-generating metabolic routes, NMNAT has the clear potential to entirely halt NAD^+^ formation if inhibited (in all of its isoforms). In turn, such complete obstruction of NAD^+^ synthesis is anticipated to heavily affect cell metabolism, including, possibly even to a higher extent, that of cancer cells. One additional advantage of NMNAT inhibition compared to either NAMPT or NAPRT blockades is that cancer cells would not be able to adapt to NMNAT inhibition through alternative NAD^+^-producing routes. In line with this notion, *NMNAT1* deletion has reduced the viability of the acute myeloid leukemia (AML) cell lines MOLM13 and OCI-AML2, which is not rescued through the supplementation of the NAD^+^ precursors NAM, NA, NR, or NMN (that conversely rescued the same cell lines from FK866) and repressed leukemia progression in two patient-derived xenograft models [94]. *NMNAT1* depletion in AML cells reduced nuclear NAD^+^, which in turn promoted p53 activity presumably by reducing SIRT6/7-mediated P53 deacetylation [94]. Ultimately, *NMNAT1* deletion induced AML cell apoptosis and sensitized AML cells to the BCL2 inhibitor, venetoclax [94]. Importantly, *NMNAT1* deletion did not impair normal hematopoiesis, suggesting that NAD^+^ biogenesis during hematopoiesis is governed by other NMNAT isoforms [94]. Although targeting NMNAT in hemato-oncology seems a viable approach and tumor resistance acquisition (at least via alternative NAD^+^ production mechanisms) would not be anticipated, multiple factors should be taken into account during the development of NMNAT inhibitors.

First, and as mentioned above, NMNAT simultaneously exists in three non-redundant isoforms inside the cell (i.e., NMNAT1, 2, and 3) [93]. These three isozymes are encoded by different gene loci and have distinct subcellular localizations and substrate affinities, as well as different oligomeric conformations (summarized in Table 2) [93,95,96]. NMNAT1 is a nuclear enzyme and functions as a homohexamer [93,95]. NMNAT2 localizes to the cytosol and to the Golgi apparatus and acts as a monomer [93,96,97]. NMNAT3 resides in the mitochondria (and probably also in the cytosol) and acts as a homotetramer [93,95,98]. NMNAT isoforms regulate cellular NAD^+^ in a compartment-specific manner [99]. Nuclear NAD^+^ levels are tightly compartmentalized and controlled by NMNAT1. NMNAT2 regulates cytosolic NAD^+^ levels. A recent study highlighted that NMNAT1 and NMNAT2 compete for their common substrate NMN during adipocyte differentiation, where the induction of NMNAT2 expression enhances cytoplasmic NAD^+^ (which is linked to improved glucose metabolism) and reciprocally reduces nuclear NAD^+^, affecting the PARP1-mediated adipogenic gene expression [100]. By contrast, NMNAT3 does not seem to play a critical role in the maintenance of NAD^+^ levels inside the mitochondria, which appear to be more reliant on the import of NAD^+^ from the cytoplasm, where NAD^+^ is synthesized by NMNAT2 [101,102]. Indeed, the depletion of NMNAT2 has been shown to reduce mitochondrial NAD^+^ concentrations in HEK293 and HeLa cells, whereas NMNAT3 depletion does not alter mitochondrial NAD^+^ levels in HeLa cells [101]. Consistently, neither a reduction in mitochondrial NAD^+^ concentrations nor impairments of glycolysis or citric acid cycle have been detected in skeletal muscle tissues of Nmnat3-KO mice [102]. In keeping with the mitochondrial uptake of cytosolic NAD^+^, SLC25A51 was recently identified as a mammalian mitochondrial NAD^+^ transporter [103,104]. A notable exception to this notion may be represented by red blood cells, which show a marked dependence on NMNAT3 for NAD^+^ formation [105,106]. Erythrocytes indeed express high NMNAT3 but weak or no NMNAT1 and NMNAT2 levels [105,106]. The genetic ablation of Nmnat3 in mature red blood cells blunts NAD^+^ levels, impairs glycolytic flow, and causes hemolytic anemia [106]. However, the degree of contribution of each NMNAT isoform to the different cellular NAD^+^ pools still requires further investigation. The different cellular localizations of the three NMNAT isoforms should thus be considered when studying NMNAT inhibitors that may target one isoform rather than the others in order to achieve the desired antitumor effects while minimizing adverse events.

Another factor that needs to be taken into consideration when thinking of NMNAT enzymes as targets for treating cancer is that human tissues display different expression levels of the three NMNAT isoforms, suggesting that the extent to which each NMNAT isoform participates in cellular NAD^+^ pool formation is likely to vary among the different tissues. In turn, this may affect how tumors arising from different tissues or organs are sensitive to such agents. NMNAT1 is the most widely expressed NMNAT isoform across a large number of human tissues/organs, including the heart, kidneys, skeletal muscles, liver, pancreas, and placenta [93,95]. Although little is known about NMNAT1 expression in tumors, analyses of cancer genomic databases using the cBioPortal platform showed that a deep deletion of *NMNAT1* has been observed in several tumor types [107]. Consistently, another study detected downregulated NMNAT1 expression in a subset of lung cancer cell lines (14 out of 36 cell lines) [108]. This reduction in NMNAT1 expression was suggested to be mediated through a heterozygous deletion of the NMNAT1 locus, which is located in a chromosomal region that is deleted in almost one-fifth of lung cancers (chromosome 1: 9746391-31610219) [108]. Under basal conditions, suppressed NMNAT1 expression levels among lung cancer cell lines have not been accompanied by proportional reductions in NAD^+^ levels, implying that the other two NMNAT isoforms are arguably the main drivers of total cellular NAD^+^ levels [108]. However, low NMNAT1-expressing lung cancer cell lines are more sensitive to the DNA-damaging agent, doxorubicin, compared to high NMNAT1-expressing lung cancer cell lines [108]. Whether doxorubicin treatment differentially affects total NAD^+^ levels in low vs. high NMNAT1-expressing lung cancer cells has not been investigated. However, these findings raise an interesting possibility in that reducing nuclear NAD^+^ levels via NMNAT1 inhibition might be sufficient to cause cancer cell death without actually affecting the total cellular NAD^+^ content [108]. In MCF-7 breast cancer cells, NMNAT1 interacts with the nuclear NAD^+^-consuming enzymes PARP1 and SIRT1 and supports their ribosylation and deacetylation activities, respectively [109,110]. NMNAT2 is also expressed in several tissues, including the brain (in abundant amounts), heart, kidneys, skeletal muscles, and pancreas [93]. The involvement of NMNAT2 in tumor development and progression has been highlighted in several cancer types. Colorectal cancer tissues show elevated NMNAT2 expression levels compared to adjacent normal tissues, and the NMNAT2 expression was found to correlate with colorectal cancer TNM staging and invasiveness [111,112]. The NMNAT2 expression was found to be positively associated with P53 expression and negatively correlated with SIRT6 levels [111,112]. These findings suggested that NMNAT2 could be involved in colorectal cancer development through a P53-mediated mechanism and that NMNAT2 expression could be enhanced through SIRT6 downregulation (although such a role for SIRT6 in NMNAT2 expression regulation remains to be confirmed) [111,112]. Similarly, NMNAT2 (and also NMNAT1) are implicated in promoting glioma proliferation by inhibiting P53-mediated apoptosis through the regulation of NAD^+^-dependent post-translational modifications of P53 [107]. NMNAT2 itself was shown to be a direct downstream target of P53, i.e., DNA damaging agents were found to induce the expression of NMNAT2 in U2-OS osteosarcoma cells through a P53-dependent mechanism [113]. Notwithstanding, this osteosarcoma cell line also showed an upregulated NMNAT1 expression in response to DNA damage [108]. Lung adenocarcinoma shows high expression levels of NMNAT2, and the expression of this gene in this type of cancer was found to negatively correlate with patient survival in two different databases [114]. In lung cancer, NMNAT2 expression was shown to be regulated by deoxyguanosine kinase (DGUOK), a critical enzyme for mitochondrial purine metabolism [114]. *DGUOK* silencing in lung adenocarcinoma cells reduces the NAD^+^ levels and downregulates NMNAT2 expression (both at the mRNA and at the protein level) through a mechanism that does not depend on mitochondria complex I activity [114]. Ovarian cancer was also shown to markedly upregulate NMNAT2 expression to enhance NAD^+^ formation and thereby promote the NAD^+^-dependent mono ADP-ribosylation (MAR) of ribosomal proteins through the catalytic activity of PARP-16 [115,116]. Ribosome MARylation, in turn, supports ovarian cancer growth by orchestrating cellular protein homeostasis mainly by inhibiting uncontrolled translation and preventing the accumulation of toxic protein aggregates inside the tumor cells [115,116]. Therefore, this study suggested that targeting NMNAT2 using inhibitors could be a promising strategy for treating ovarian cancer [115,116]. Examining *NMNAT2* gene alterations in the cBioPortal database revealed that the *NMNAT2* gene is also amplified in tumors such as cholangiocarcinoma, invasive breast carcinoma, and hepatocellular carcinoma [107]. Taken together, these studies underscore the pro-oncogenic roles of NMNAT1 and NMNAT2 in several solid tumors, implying that particularly these two NMNAT isoforms could represent promising therapeutic targets. NMNAT3 is mainly expressed in the spleen and in red blood cells but very little is known about its involvement in cancer.

The last point that should be carefully considered is the toxicity that might arise with NMNAT inhibitors. NMNAT isoforms play fundamental roles in early development as well as in preserving neuronal integrity and protecting against neurodegeneration [95]. NMNAT has also been shown to display its neuroprotective effects by acting as a molecular chaperone independent of its enzymatic function in NAD^+^ biosynthesis [117,118,119,120]. Several mechanisms are associated with the neuroprotective effect of NMNATs, including the suppression of ROS production, mitochondrial stabilization [120], promotion of the clearance of hyperphosphorylated Tau protein oligomers [121], autophagic clearance of amyloid plaques in Alzheimer’s disease models [122], and reduced accumulation of mutant Huntingtin (Htt) aggregation [123]. Moreover, NMNAT1 is important in retinal development and physiology, and NMNAT1 mutations are linked to the occurrence of detrimental retinal degenerative conditions such as Leber congenital amaurosis [124,125,126,127]. *Nmnat1* gene therapy has shown protective effects against glaucomatous neurodegeneration and *Nmnat1*-associated retinal degeneration in mice models [128,129]. NMNAT1 was also reported to be a stress response protein, as its expression was induced after exposure to hypoxia, heat shock, and oxidative stress [130].

Similar to NMNAT1, NMNAT2 is also a recognized neuroprotective factor that plays an essential role in maintaining axonal integrity through its NAD^+^ synthetase catalytic activity as well as by acting as a molecular chaperone [131]. The suppression of NMNAT2 skews the homeostatic balance that is maintained by the NMNAT2–NAD^+^–SARM1 axis and results in axonal degeneration, an effect that is commonly observed in many neurodegenerative diseases or as a physiological response to nerve injury (Wallerian degeneration) [131,132]. NMNAT2 also protects against chemotherapy-induced peripheral neuropathy [131]. A recent study illustrated that peripheral neuropathy caused by the chemotherapeutic agents vincristine and bortezomib is also triggered through axonal NMNAT2 depletion and that the consequent NAD^+^ loss is induced through SARM1 activation [133]. The molecular mechanism involves the destruction of the short-lived NMNAT2 in axons, which results in an increased NMN/NAD^+^ ratio [134]. The NAD^+^-consuming enzyme SARM1, which can sense an elevated NMN/NAD^+^ ratio, is activated and further degrades NAD^+^, ultimately triggering a programmed axonal self-destructive program [134]. In line with the notion that skewed NMN/NAD^+^ can cause peripheral neuropathy, NAMPT inhibition using the novel agent A4276 was found to protect against Wallerian degeneration (during which NMNAT2 is downregulated) and peripheral neuropathy induced by vincristine and paclitaxel [74]. Interestingly, paclitaxel-induced peripheral neuropathy has also been reversed using NAMPT activators [135]. Both NAMPT inhibitors and activators presumably exert these protective effects through the favorable modulation of the NMN/NAD^+^ ratio (by reducing NMN production in the case of inhibitors or replenishing NAD^+^ levels in the case of activators) and thereby hampering SARM1 activation [74,135]. Overall, given its potential to skew the NMN/NAD^+^ ratio, treatment with NMNAT inhibitors may cause side effects, such as neuropathies, and clinical studies on such molecules should carefully monitor these aspects.

### 4.2. NMNAT Inhibitors

The development of potent and selective NMNAT inhibitors and their application in cancer therapy is still in its infancy. On one hand, targeting microbial NMNAT has already gathered considerable attention in the recent past from the perspective of developing antibiotics and antimalarial drugs. Indeed, the crystal structures of a large number of bacterial and protozoal NMN/NAMN adenylyltransferases have been already revealed [136,137,138,139,140,141]. Studies comparing NMNAT structures from different organisms revealed that bacterial NMN/NAMN adenylyltransferases (known as NadDs) from different species are structurally close but display significant differences in comparison to their human counterparts [142,143,144]. Thus, inhibitors can be developed to selectively target the bacterial/protozoal NMNAT while sparing the human NMNAT enzyme. On the other hand, these findings hint that the available bacterial NadD inhibitors would not presumably inhibit the human NMNAT enzyme, making their usefulness as therapeutic agents against human diseases, such as cancer, very doubtful.

The primary sequence of human NMNAT1 was reported in the early 2000s, and a human recombinant NMNAT1 enzyme was characterized in the same study [145]. The 3D crystal structures of human NMNAT1 and NMNAT3 isoforms were resolved later in both the apo-form and in complex with substrates [98,143,146,147]. By contrast, only a homology-based structural NMNAT2 model is reported and the actual crystal structure of NMNAT2 is yet to be resolved [148]. Therefore, utilizing in silico drug design as a tool to discover selective NMNAT2 inhibitors is envisaged to encounter additional hurdles compared to the two other isoforms. Overall, developing inhibitors that can effectively suppress the enzymatic activity of human NMNAT isoforms is expected to be a challenging process. In the next section, we will summarize the efforts that have been undertaken in this field.

As of today, a very limited number of NMNAT inhibitors has been identified (summarized in Table 3) and only one compound called Vacor (which was originally used as a rodenticide) was reported to exhibit antitumor effects that were mediated through NMNAT2 inhibition [149]. Vacor is a metabolite that undergoes the same NAD^+^-biosynthetic cycle as NAM, being converted into Vacor mononucleotide (VMN). Afterward, NMNAT2 transforms VMN into Vacor adenine dinucleotide (VAD) (which is analogous to NMN’s transformation into NAD^+^) [149]. VAD is a toxic metabolite that blocks the activities of NAMPT, NMNAT, and other NAD^+^-dependent enzymes, leading to a catastrophic NAD^+^ depletion, metabolic impairment, and cancer cell lethality [149]. NMN (1 mM) was shown to be able to rescue NAD^+^ levels in SH-SY5Y neuroblastoma cells from FK866 and GMX1778 but not from Vacor [149]. Vacor showed marked in vivo antitumor activity in NMNAT2-expressing melanoma and neuroblastoma xenografts [149]. The antitumor activity of Vacor was exclusively limited to NMNAT2-expressing tumors, while tumor cell lines that showed no NMNAT2 expression were entirely resistant to Vacor [149]. VMN could also be produced through the Vacor analog Vacor riboside (VR) through the activity of nicotinamide riboside kinases [150]. However, whether VR also exhibits anticancer activity against NMNAT2-positive tumors needs to be determined.

Gallotannin is the most well-known pan NMNAT inhibitor with different potencies toward the three human isozymes: it showed an IC50 of 2 µm against NMNAT3, 10 µM against NMNAT1, and 55 µm against NMNAT2 [93]. However, NMNAT is not the only target of gallotannin, which can also inhibit other enzymes, such as poly(ADP-ribose) glycohydrolase [151]. By contrast, the tannin derivative, epigallocatechin gallate (EGCG), activates the NMNAT human isozymes with different degrees of enhancement in enzyme activity [93]. The nucleotides Np_3_AD, Np_4_AD, and Nap_4_AD (which are NAD^+^ analogs that contain oligophosphate groups) can also effectively inhibit the three human NMNAT isoforms [152]. These compounds are geometric multi-substrate NMNAT inhibitors with inhibition constants in the micromolar range [152]. Np_3_AD and Np_4_AD inhibit NMNAT2 more effectively than Nap_4_AD, which has shown a more pronounced inhibition toward NMNAT1 and 3 [152]. N-2′-MeAD and Na-2′-MeAD are other NAD^+^ analogs that can selectively inhibit the NMNAT3 isoform with IC50 values of 0.19 mM and 1.12 mM, respectively [153]. The anticancer activities of these NAD^+^ analogs remain to be elucidated. Recently, several positive and negative NMNAT2 modulators were identified by screening 1280 compounds using a Meso Scale Discovery (MSD)-based screening platform [154]. Caffeine was found to positively modulate NMNAT2 expression and to protect neurons against vincristine-induced cell death [154]. By contrast, the negative NMNAT2 modulators, cantharidin, retinoic acid, and wortmannin reduced NMNAT2 levels in neurons and exacerbated vincristine-induced neuronal cell death [154]. High-throughput screening of oncology libraries containing a total of 912 compounds for the NMNAT1-catalyzed reaction led to the identification of 2,3-Dibromo-1,4-naphthoquinone (DBNQ) as a potent NMNAT1 inhibitor [155]. DBNQ was found to compete with both reaction substrates NMN and ATP and to inhibit both the forward and reverse reactions with IC50 values of 0.76 and 0.26 µM, respectively [155]. The activities of DBNQ in NMNAT1-expressing cells (including cancer cells) as well as on the two other isozymes are yet to be established. Notably, the bioluminescent NMNAT assay that was devised and employed in that study provided a starting platform for identifying and characterizing additional NMNAT modulatory chemotypes [155].

Apart from being a target for inhibition, NMNAT could also be harnessed in cancer treatment by mediating the activation of prodrugs into their cytotoxic metabolites [156]. For instance, the prodrug, tiazofurin, can be phosphorylated into tiazofurin 5-monophosphate (TRMP). Thereafter, NMNAT catalyzes the transformation of TRMP into its active metabolite, thiazole-4-carboxamide adenine dinucleotide (TAD) [157]. TAD is a NAD^+^ analog and an excellent inhibitor of inosine 5-monophosphate dehydrogenase (IMPDH), a key enzyme in guanylate nucleotide synthesis [156]. Indeed, disrupting guanylyl synthesis is fatal to cancer cells. Colorectal cancer cells with a low NMNAT2 expression have demonstrated resistance to tiazofurin, which could be reversed by overexpressing NMNAT2 in tiazofurin-resistant colorectal cancer cell lines [158]. Likewise, NMNAT is also required for the metabolic activation of similar anticancer agents, such as selenazofurin and benzamide riboside [159].

As mentioned earlier, in bacteria and in *P. falciparum*, several NadD/NMNAT inhibitors have been identified (with the goal of using them as antibiotics and/or antimalarial agents). Specifically, most of these inhibitors were discovered through studies on *E. coli*, *B. anthracis*, and *P. falciparum*, and the best compounds were found to have IC50 values in the low micromolar range [142,160,161,162,163,164] (summarized in Table 4). It is worth mentioning that NadD inhibitors compounds 3_02 and 1_02 (Table 4) failed to inhibit the activity of any human NMNAT homolog [142,160]. In addition, no information about their antitumor effects is currently available. Future studies should possibly study these molecules that may offer meaningful cues for developing inhibitors against mammalian NMNATs.

**Table 2 ijms-25-02092-t002:** Comparison between the three isoforms of human NMNAT.

Point of Comparison	NMNAT1	NMNAT2	NMNAT3
Cellular location	Nucleus [93]	Cytoplasm and Golgi (anchored to the cytosolic face of the Golgi through palmitoylation) [93,96]	Mitochondria mainly, but also in the cytosol [93,98] The location differs according to the cell type
Chromosomal location	Chromosome 1 p36.22	Chromosome 1 q25.3	Chromosome 3q23
Tissue expression (reported as either mRNA or protein)	Heart, skeletal muscles, kidney, liver, pancreas, and placenta (most abundant form) [93]	Brain, heart, skeletal muscles, and pancreas [93]	Spleen, lung, and kidney [93] Red blood cells [105]
Structure	Homohexamer	Monomer	Homotetramer
	Crystal structure solved in the ligand-free form and in complex with NMN, NAD^+^, and NAAD [143,146,147]	The 3D structure is not solved [97,148]	Crystal structure solved in the ligand-free form and in complex with NMN, ATP, and NAD^+^ [98]
Enzymatic reaction, kinetic parameters, and substrate affinity	ATP binds before NMN [152]	ATP binds before NMN [152]	NMN binds before ATP [152]
	High affinity for NMN and ATP (Km NMN = 34 µM and Km ATP = 40 µM) [93]	Lower affinity for ATP (Km NMN = 32 µM and Km ATP = 204 µM) [93]	Lower affinity for NMN (Km NMN = 209 µM and Km ATP = 29 µM) [93]
	Similar preference for NMN and NAMN [93]	Prefers NMN over NAMN [93]	Similar preference for NMN and NAMN [93]
	Similar preference for NMN and NMNH (NAD^+^/NADH synthesis is 1.2) [93]	Similar preference for NMN and NMNH (NAD^+^/NADH synthesis is 1.1) [93]	Prefers NMNH over NMN (NAD^+^/NADH synthesis is 0.5) [93]

**Table 3 ijms-25-02092-t003:** Effects of NMNAT modulators on human NMNAT1-3.

Modulator	NMNAT1	NMNAT2	NMNAT3
Gallotannin	Inhibitor (IC50 = 10 µM) [93]	Inhibitor (IC50 = 55 µM) [93]	Inhibitor (IC50 = 2 µM) [93]
Np_3_AD	Inhibitor (Ki is 89 µM toward NMN and 56.3 µM toward ATP) [152]	Inhibitor (Ki is 31.5 µM toward NMN and 35.9 µM toward ATP) [152]	Inhibitor (Ki is 66.8 µM toward NMN and 40.6 µM toward ATP) [152]
Np_4_AD	Inhibitor (Ki is 31.1 µM toward NMN and 49.2 µM toward ATP) [152]	Inhibitor (Ki is 25.8 µM toward NMN and 24.2 µM toward ATP) [152]	Inhibitor (Ki is 73.6 µM toward NMN and 29.8 µM toward ATP) [152]
Nap_4_AD	Inhibitor (Ki is 67.9 µM toward NMN and 59.1 µM toward ATP) [152]	Inhibitor (Ki is 328.3 µM toward NMN and 174.5 µM toward ATP) [152]	Inhibitor (Ki is 88.3 µM toward NMN and 32.8 µM toward ATP) [152]
Na-2′-MeAD	Weak inhibitor (28% inhibition at 1 mM) [153]	Weak inhibitor (33% inhibition at 1 mM) [153]	Inhibitor (IC50 = 1120 µM) (81% inhibition at 1 mM) [153]
N-2′-MeAD	Very Weak inhibitor (9% inhibition at 1 mM) [153]	Very Weak inhibitor (9% inhibition at 1 mM) [153]	Inhibitor (IC50 = 190 µM) (65% inhibition at 1 mM) [153]
DBNQ	Inhibitor (IC50 is 0.76 µM for the forward reaction, and 0.26 µM for the reverse reaction) [155]	N/A	N/A
EGCG	Activator (1.2 fold activation at 50 µM) [93]	Activator (2.28 fold activation at 50 µM) [93]	Activator (1.42 fold activation at 50 µM) [93]

N/A; Not available.

**Table 4 ijms-25-02092-t004:** Summary of selected bacterial nicotinate mononucleotide adenylyltransferase (NadD) and NAD^+^ synthetase (NadE) inhibitors and their reported activities.

Name and Structure	Enzyme Inhibition (IC50/Ki)	Antibacterial/Antiparasitic Activity (MIC)	Antitumor Activity
Compound **3_02** 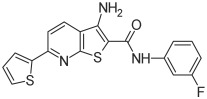	Inhibits nicotinate mononucleotide adenylyltransferases of *E. coli* (IC50 = 65 µM, Ki = 25 ± 9 µM toward NAMN and 21 ± 9 µM toward ATP) and *B. anthracis* (IC50 = 36 µM, Ki = 18 ± 4 µM toward NAMN and 32 ± 5 µM toward ATP) but not human NMNAT1-3 [142]	Reduces the growth of *E. coli* (MIC50 = 160 µM), *B. anthracis* and *B. subtilis* (MIC50 = 80 µM) [142]	N/A
Compound **1_02** 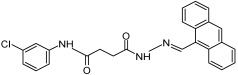	Inhibits nicotinate mononucleotide adenylyltransferases of *E. coli* (IC50 = 15 µM, Ki = 8 ± 3 µM toward NAMN and 5 ± 1 µM toward ATP) and *B. anthracis* (IC50 = 25 µM, Ki = 9 ± 3 µM toward NAMN and 10 ± 2 µM toward ATP) but not human NMNAT1-3 [142]	Reduces the growth of *E. coli* (MIC50> 80 µM) [142]	N/A
Compound **1_03** 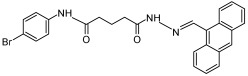	Inhibits nicotinate mononucleotide adenylyltransferases of *E. coli* (IC50 = 18 µM), *B. anthracis* (IC50 = 33 µM) [142], and *P. falciparum* [162], but not human NMNAT1-3 [142]	Reduces the growth of *E. coli* (MIC50> 80 µM), *B. anthracis* (MIC50 = 15 µM) and *B. subtilis* (MIC50 = 10 µM) [142]Suppresses the growth of the *P. falciparum* parasite (MIC50 = 50 µM [163] and 8.09 ± 5.11 µM [162]) and arrests its growth inside the erythrocytes at the trophozoite stage (at 100 µM) [163]	N/A
Compound **N2-11** 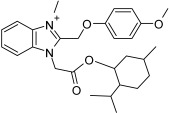	Inhibits nicotinate mononucleotide adenylyltransferases of *M. tuberculosis* (IC50 = 6 ± 1 µM) [164]	Reduces the growth of *M. smegmatis* (MIC = 5 µM), and *M. tuberculosis* (MIC = 28.2 µM) [164]	N/A
Compound **5824** 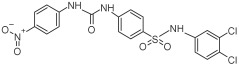	Inhibits NAD^+^ synthetase of *B. anthracis* (IC50 = 6.4 µM and 10 µM) [161,165]Inhibits nicotinate mononucleotide adenylyltransferases of *B. anthracis* (IC50 = 2 µM) [165]Inhibits the activity of human NAD^+^ synthetase enzyme (NADSYN1) in a dose-dependent manner (i.e., at 1, 2, and 3 µM) [69]	Reduces the growth of *B. anthracis* (MIC = 1.9 µM in LB media and 2.8 µM in MH media) [165]Reduces the viability of *L. donovani* promastigotes at micromolar concentrations [166]	Reduces NAD^+^ levels and the viability of several PH-amplified cancer cell lines but not normal cells [69]Represses the growth of PH-amplified OV4 ovarian cancer tumors in vivo [69]
Compound **5{1}** 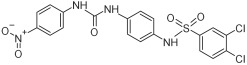	Inhibits NAD^+^ synthetase of *B. anthracis* (IC50 = 32.7 µM) [161]Inhibits nicotinate mononucleotide adenylyltransferase of *B. anthracis* (IC50 = 7.3 µM) [161]	Reduces the growth of *B. anthracis* (MIC = 2.8 µM in LB media and 3.8 µM in MH media) [161]	N/A
Compound **VIB8E** 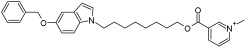	Inhibits NAD^+^ synthetase of *B. subtilis* (IC50 = 20 µM) [167]	Reduces the growth of gram-positive bacteria such as *B. subtilis* (MIC = 6.2 µM), *S. aureus* (MIC = 1.5 µM), and methicillin-resistant *S. aureus* (MIC is between 3.1 and 12.5 µM), but not gram-negative bacteria such as *S. enteritidis* and *P. aeruginosa* [167]	N/A
Compound **VD1** 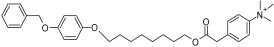	Inhibits NAD^+^ synthetase of *B. subtilis* (IC50 = 20 µM) [168]	Reduces the growth of gram-positive bacteria such as *B. subtilis* (MIC = 2 µg/mL), *S. aureus* (MIC = 9.4 µg/mL), and methicillin-resistant *S. aureus* (MIC = 3.1 and 6.2 µg/mL), but not gram-negative bacteria such as *S. enteritidis* and *P. aeruginosa* [168]	N/A
Compound **2A** 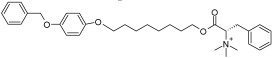	Inhibits NAD^+^ synthetase of *B. subtilis* (IC50 = 22 ± 2.4 µM) [169]	Reduces the growth of gram-positive bacteria such as *B. subtilis*, *S. aureus*, and methicillin-resistant *S. aureus* (MIC = 0.78 µM for all), but not *P. aeruginosa* [169]	N/A
Compound **4f** 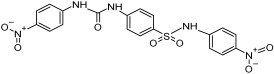	Inhibits NAD^+^ synthetase of *M. tuberculosis* (IC50 = 90 ± 5 µM) [170]	Reduces the growth of *M. tuberculosis* (MIC = 37 µg/mL in rich media and 25 µg/mL in minimal media) [170]	N/A

N/A; Not available.

## 5. Targeting NAD^+^ Synthetase

Mammalian NAD^+^ synthetase (NADSYN1) is an amidotransferase enzyme that is responsible for the third and last step of NAD^+^ biogenesis in the PH pathway. In a two-step reaction that requires Mg^+2^, NADSYN1 catalyzes the ATP-dependent conversion of NAAD into NAD^+^. The amino acid glutamine serves as the nitrogen source for mammalian NAD^+^ synthetase (while the bacterial NAD^+^ synthetase (NadE) can use either ammonia or glutamine as a nitrogen donor depending on the bacterial species) [171,172]. A second isoform of the human NAD^+^ synthase enzyme was initially reported. Its peculiarity was that it was reportedly able to use ammonia, instead of glutamine, as a nitrogen source. However, this enzyme, which was named NADSYN2, was later found to be a pseudomonal NAD^+^ synthetase [172,173]. A northern blot analysis of mouse tissues revealed that the *NADSYN1* gene is abundantly expressed in the small intestine, kidney, liver, and testis, whereas skeletal muscle and heart show a very weak *NADSYN1* expression [171]. Despite the occurrence of redundant NAD^+^ biosynthetic routes in humans, NADSYN1 seems fundamental for normal human physiology. Several case reports identified patients with biallelic pathogenic variants of the *NADSYN1* gene, which were found to be associated with a rare, severely debilitating, and potentially lethal condition called congenital NAD^+^ deficiency disorder [174,175,176,177,178]. Patients suffering from this condition present with severe cardiac, limb, vertebral, and renal defects [174,175,176,177,178]. *NADSYN1* is amplified in many solid tumors, such as lung, bladder, and breast cancer (Figure 3) [69]. Esophageal and head and neck cancers show the most striking *NADSYN1* amplification frequency, (i.e., 19.23% and 13.77% of these tumors have amplified *NADSYN1* gene expressions, respectively), implying that the PH pathway is heavily implicated in the pathophysiology of these two malignancies (Figure 3). In line with this notion, elevated expression levels of NAPRT, the first enzyme in the PH pathway, were also found to occur in these two cancer types to support NAD^+^ biosynthesis [70,88,179]. NAPRT expression was associated with resistance of head and neck tumors to NAMPT inhibitors and with increased risk of the development of esophageal precancerous lesions [88,179].

So far, similar to NMNAT, NAD^+^ synthetase has mostly been explored as a therapeutic target to treat infectious diseases. Thus, the goal has been to find inhibitors of microbial NAD^+^ synthetase, rather than of human NADSYN1. In this area, several inhibitors of bacterial NadE have been identified, with the best ones showing IC50 values in the low micromolar range [161,165,166,167,168,169,170] (summarized in Table 4). One of these inhibitors of bacterial NadE (i.e., compound 5284) was studied by Chowdhry and colleagues, who investigated its ability to inhibit the human NADSYN1 enzyme [69]. Indeed, compound 5284 dose-dependently reduced the enzymatic activity of the purified human NADSYN1 enzyme [69]. It markedly blunted NAD^+^ levels and reduced the viability of several PH-amplified cancer cell lines (while not affecting normal cells). Similar effects were also achieved in xenograft-bearing mice, where treatment with compound 5248 suppressed the growth of PH-amplified OV4 ovarian cancer tumors and hampered NAD^+^ production inside these tumors [69]. Noteworthy, this is the first and only study providing experimental evidence of anticancer activity attributed to NAD^+^ synthetase inhibitors reflecting the need to identify more NADSYN1 inhibitors [69]. At the time of that study, the crystal structure of the human NADSYN1 enzyme was not known. However, it was resolved one year later [180]. Notably, resolving the crystal structures of the NadE enzyme from several bacterial species, particularly *B.subtilis* and *B.anthracis*, was pivotal in the discovery of microbial NAD^+^ synthetase inhibitors [181,182,183]. Thus, the availability of the crystal structure of the human NADSYN1 enzyme is expected to pave the way for subsequent studies that aim at identifying novel chemotypes that specifically target human NADSYN1, particularly for their potential exploitation in oncology.

## 6. Alternative NAD^+^ Precursors and NAD^+^ Biosynthetic Enzymes

While NAM and NA play the major roles in keeping proper NAD^+^ levels in healthy and neoplastic tissues via the salvage and the PH metabolic pathways, respectively, the contribution of other NAD^+^ precursors, such as tryptophan and NR, as well as our intestinal flora in supporting the NAD^+^ metabolism must not be overlooked. As aforementioned, tryptophan catabolism generates NAD^+^ through the more complex kynurenine (de novo) pathway, which involves a chain of multiple enzymatic steps to yield QA, which then converges into the PH pathway upon its transformation to NAMN through QPRT. Although this pathway is not active in most bodily tissues, in vivo radiolabeling and tracing studies demonstrated that the liver predominantly relies on tryptophan to produce NAD^+^ and that hepatic NAD^+^ degradation generates NAM, which is released into the circulation and utilized by other tissues to form NAD^+^ [26]. Multiple enzymes in the de novo pathway are interesting therapeutic targets for curing cancer. For instance, indoleamine 2,3-dioxygenase (IDO1), the first enzyme in this pathway that drives the catabolism of tryptophan into kynurenine, is expressed in numerous tumors and its expression is associated with poor prognosis [184,185]. IDO1 inhibitors have shown robust antitumor activity and are currently assessed in clinical trials as single agents or in combination with cancer immunotherapies (reviewed in [186,187]). IDO1 inhibitors prevent the tryptophan deprivation and accumulation of kynurenine (and also its metabolites) in the cells and in the tumor microenvironment. This, in turn, is a recognized mechanism of tumor immune evasion, promoting the activity of regulatory T-cells and thereby hampering the activity of effector T-cells. Whether intra-tumor NAD^+^ depletion through IDO1 inhibitors might be an additional mechanism underlying the antitumor activity of these agents remains to be defined. In support of this notion, first-in-class dual NAMPT-IDO1 inhibitors were recently discovered and showed potent antiproliferative and antimigration effects in lung cancer cells [188]. The most promising NAMPT-IDO1 inhibitor compound, 10e (IC50 = 57.7 nM against recombinant NAMPT and 160 nM against IDO1 in HeLa cells), significantly suppressed the growth of xenografted A549/R cells (an FK866 and taxol-resistant lung cancer cell line) as a single agent and also markedly sensitized the tumors to taxol [188]. Inhibiting the activity of alpha-amino-beta-carboxy-muconate-semialdehyde decarboxylase (ACMSD), another gatekeeping enzyme in the kynurenine pathway that diverts tryptophan catabolism away from NAD^+^ biosynthesis, improving the mitochondrial function by increasing NAD^+^ availability, whereas its overexpression in transgenic mice models rendered them addicted to dietary NA to produce NAD^+^ and led to NAD^+^ deficiency upon dietary niacin deprivation [189,190]. Lastly, high expression levels of QPRT, the rate-limiting enzyme that converts QA to NAMN, enhance the migration and invasive properties of breast cancer cells and are associated with worsened prognoses and clinical outcomes in breast cancer patients [191,192,193]. In addition, an elevated QPRT expression in the GMX1778-resistant HT1080 fibrosarcoma cell line and in the FK866-resistant CCRF-CEM leukemia cell line was the underlying resistance mechanism to these NAMPT inhibitors (through the activation of de novo NAD^+^ biosynthesis) [194,195]. Consistent with these studies, QPRT-expressing glioma cells were shown to be resistant to concomitant oxidative stress and NAMPT inhibition through the activation of de novo NAD^+^ synthesis starting from QA [196]. Interestingly, although glioma cells can’t synthesize NAD^+^ starting from tryptophan (because they lack the de novo pathway enzyme *3*-Hydroxyanthranilate *3*,4-Dioxygenase (3-HAO)), they can still take advantage of the de novo NAD^+^-production route by utilizing the tryptophan metabolite QA, which is supplied by infiltrating microglial cells [196]. The other NAD^+^ precursor, NR, could boost NAD^+^ levels via the NMRK-mediated nucleoside pathway, which is of significant relevance in skeletal muscles [197,198,199]. The NR-NMRK pathway can mediate resistance to NAMPT inhibition in salvage-dependent tumor cells, which could be reversed via dual NAMPT and NMRK inhibition [69]. Consistent with these findings, the pharmacological inhibition or downregulation of CD73 (which mediates the extracellular conversion of NMN to NR) largely abrogates the rescue effects of NMN in FK866-treated OVCAR-3 ovarian cancer cells, A549 lung cancer cells, and U87 glioblastoma cells by impairing the extracellular provision of NR from NMN [32,200]. To our knowledge, phthalic acid is the only available inhibitor to the human QPRT enzyme, and no NMRK inhibitors have been reported so far [201].

The commensal bacteria residing in the intestinal lumen significantly influence the mammalian NAD^+^ metabolism. Through the catalytic activity of their microbial nicotinamidase (PncA) enzyme, the intestinal flora can convert NAM to NA, which, in turn, can circulate and stimulate NAD^+^ biosynthesis in mammalian host tissues (presumably also including neoplastic tissues) that possess a functional PH pathway [202]. This finding also has therapeutic implications for treatment with NAD^+^-lowering agents. In mice models of leukemia, gut flora caused tumor resistance to FK866 when mice were fed with NAM-enriched diets through the production of NA, and this resistance was reversed through antibiotic treatment that depleted the intestinal microbiota [203]. Similar results were also reported in a colorectal cancer xenograft mice model [202]. Interestingly, gut microbiota can utilize circulating NAM (i.e., NAM produced from the mammalian host tissues most commonly as a byproduct of NAD^+^-degradation and secreted into the circulation) to generate NA [204]. Moreover, oral NR was found to boost NAD^+^ levels not only via the NMRK pathway but also through its degradation first into NAM by the enzyme CD157 (also known as BST1 which is present in the small intestine) and then into NA by the gut flora [205]. Finally, NA promotes tissue NAD^+^ biosynthesis via the PH pathway [205]. Notably, NR can also be converted to NAR through the base exchange activity of BST1 [205]. Taken together, these results emphasize that NAM, besides being the precursor of the NAM salvage pathway, can also fuel the activation of the PH pathway in the presence of gut flora that typically converts it into NA.

Finally, the NAM-metabolizing enzyme NNMT can strongly impact cellular NAD^+^ homeostasis [37]. By catalyzing the methylation of NAM into MNAM (which is eventually subjected to further metabolism and/or urinary excretion), NNMT can reduce the free NAM pool and prevent NAM from entering the salvage pathway to rebuild NAD^+^ [37]. Indeed, overexpression of NNMT in SW480 colorectal cancer cells reduces NAD^+^ levels, whereas an NNMT downregulation in HT29 cells elevates cellular NAD^+^ content [206]. Consistently, the treatment of adipocytes with an NNMT inhibitor has increased NAD^+^ levels [207]. To methylate NAM, NNMT consumes methyl units from S-adenosyl-L-methionine (SAM), yielding S-adenosyl-homocysteine (SAH). Given that SAM serves as a universal methyl donor for many methyltransferases including histone methyltransferases, NNMT expression in tumors can remodel their cellular epigenetic landscapes by skewing SAM/SAH levels and by creating a state of hypomethylated histones [208]. NNMT was also reported to regulate the expression/activity of SIRT1 in prostate and breast cancer cells by supporting SIRT1 stabilization [209,210]. These findings suggest that NNMT can also influence cellular epigenetics in neoplastic cells by modulating SIRT1-mediated histone deacetylation. Notably, the upregulation of NNMT has been reported in numerous neoplasms including gastrointestinal cancers, urological cancers, skin cancers, and head and neck tumors, and has been proposed as a tumor biomarker (comprehensively reviewed in [211,212,213,214]). NNMT expression was implicated in driving tumorigenesis and aggressiveness since multiple pro-oncogenic effects were ascribed to NNMT, including the inhibition of apoptosis, promotion of cancer cell viability, cell cycle progression, and invasiveness and migration, as well as the reduction of ROS generation [206,209,215,216,217]. Furthermore, NNMT expression was associated with breast cancer resistance to adriamycin and paclitaxel, colorectal cancer resistance to 5-fluorouracil (5-FU), and melanoma resistance to dacarbazine [210,217,218]. NNMT downregulation enhances cancer cells’ sensitivity to these chemotherapeutics [210,217,218]. In light of these insights, targeting NNMT with chemical inhibitors is receiving increasing attention in cancer therapy. Notably, a considerable number of NNMT inhibitors have been recently annotated within the scope of their employment as novel anticancer agents [219,220,221]. Whether and how NNMT inhibitors affect the antitumor activity of NAD^+^-lowering drugs, including NAMPT inhibitors, remains to be determined.

## 7. Conclusions and Perspectives

Mounting evidence shows that multiple types of tumors exploit the PH pathway, particularly upon NAMPT inhibition, to meet their NAD^+^ requirements. Hence, the three enzymes that govern NAD^+^ production through this pathway (i.e., NAPRT, NMNAT, and NADSYN1) represent a set of promising targets for cancer therapy. The inhibition of enzymes that mediate the early rate-limiting steps of NAD^+^ generation (i.e., NAMPT and NAPRT) is a reasonable intervention. However, in tumors that express both NAMPT and NAPRT, blocking the activity of one enzyme will be presumably bypassed by the utilization of the other enzyme, and thus both enzymes should be concomitantly targeted. On the other hand, blocking the activity of enzymes that control the later steps in NAD^+^ generation is expected to prevent NAD^+^ generation from multiple routes/precursors at the same time. This, in turn, could achieve a more durable intratumor NAD^+^ depletion. For instance, NADSYN1 inhibition would block NAD^+^ generation from NA (through the PH pathway), NAR (through the nucleoside pathway), tryptophan, and all the kynurenine metabolites (through the de novo pathway). The fact that NMNAT plays a role in NAD^+^ synthesis through all pathways implies that its blockade could, in principle, also halt NAD^+^ biosynthesis from all possible routes. Very few inhibitors of mammalian NADSYN1 and NMNAT enzymes are available. On the other hand, NADSYN1 and NMNAT have been largely investigated as targets for developing antibacterial/antiparasitic drugs, raising the possibility that at least some of these agents with the ability to also obstruct the human enzymes could be used for treating cancer too. Human and bacterial NADSYN enzymes seem to be more similar as compared to human vs. bacterial NMNAT enzymes (at least in the substrate binding sites). A bacterial NADSYN inhibitor was indeed able to inhibit the functionally equivalent human enzyme, showing anticancer effects against PH pathway-dependent tumors [69]. On the other hand, the substantial differences that exist between human and bacterial NMNAT enzymes make exploiting the current bacterial NMNAT inhibitors in human pathologies extremely challenging. Furthermore, several aspects should be taken into account while developing human NMNAT inhibitors. These include (i) its existence in three non-redundant isoforms inside the cell (where each isoform displays unique characteristics); (ii) the variable expression levels of the three isoforms in normal tissues as well as in tumors; and (iii) the central roles that NMNAT plays in neuronal and retinal physiology, which warrants the close monitoring of neurological and retinal toxicities that might potentially arise upon treatment with NMNAT inhibitors. Another way to exploit the enzymatic activity of NMNAT enzymes consists in the administration of prodrugs that are transformed by NMNAT into cytotoxic metabolites, which then display antitumor effects through diverse mechanisms. This is the case, for instance, of anticancer agents such as Vacor, tiazofurin, and selenazofurin. Last, but not least, it is important to note that the potency of the reported inhibitors against all three PH pathway enzymes lies within the micromolar-to-millimolar range. Moreover, a limited number of these inhibitors have been associated with an antitumor activity either alone (as in the case of the NADSYN1 inhibitor compound 5284) or in combination with NAMPT inhibitors (which is the case with the NAPRT inhibitor 2-HNA). Therefore, overall, there remains a crucial need to develop more potent and selective NAPRT, NMNAT, and NADSYN1 inhibitors with optimized drug-like properties. The anticancer activity of these inhibitors should be addressed in preclinical and then in clinical studies. Finally, reducing the availability of dietary and microbiota-derived NA can also hamper the ability of tumors to utilize the PH pathway to build NAD^+^ and hence maximize the efficacy of NAD^+^-depleting agents.

## Figures and Tables

**Figure 1 ijms-25-02092-f001:**
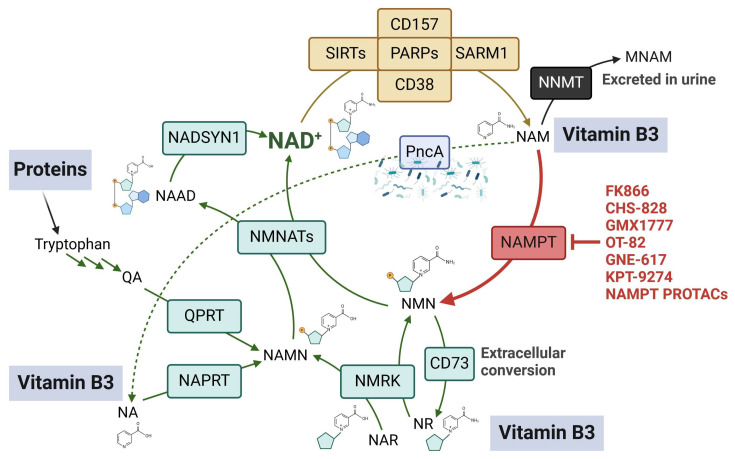
Overview of NAD^+^-biosynthetic routes. Green arrows represent the pathways that could be enabled in response to the inhibition of the predominant salvage pathway (red arrow) by NAMPT inhibitors. The yellow arrow represents NAD^+^ consumption through diverse NAD^+^-degrading enzymes. This figure was created with BioRender.com (accessed on 28 January 2024).

**Figure 2 ijms-25-02092-f002:**
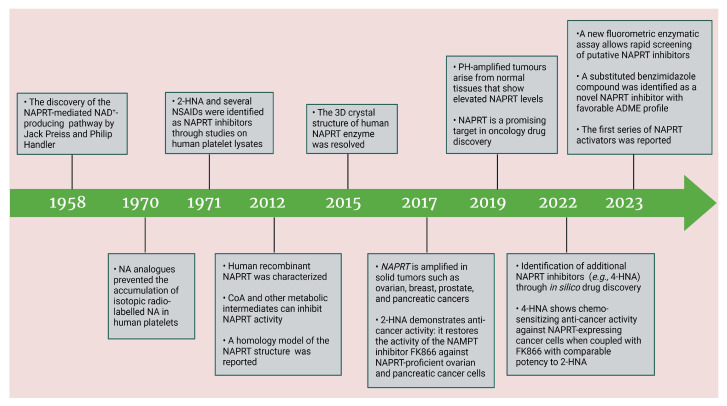
Historic overview of the milestone discoveries regarding NAPRT and its inhibitors and their exploitation in cancer. The figure was created with BioRender.com (accessed on 28 January 2024).

**Figure 3 ijms-25-02092-f003:**
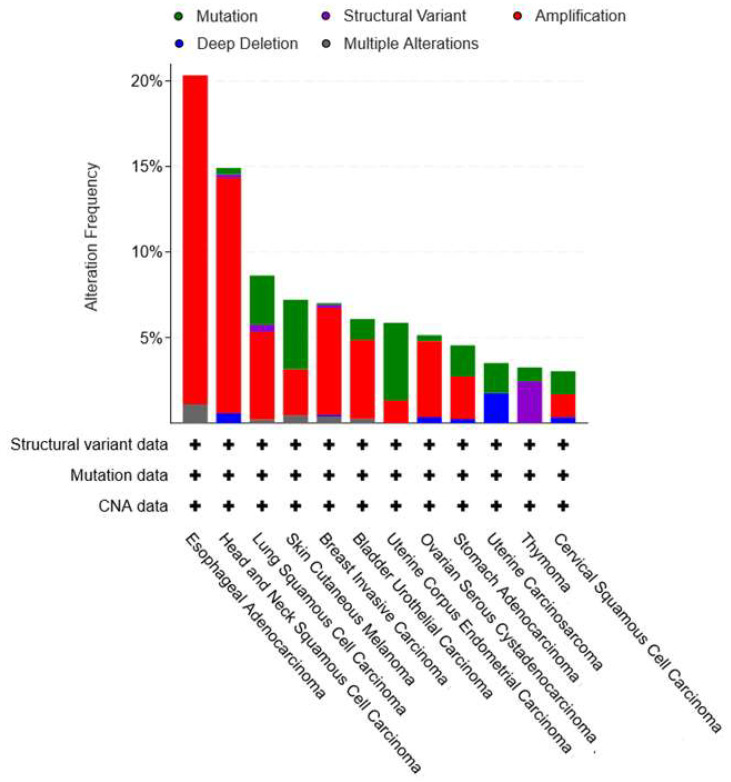
NADSYN1 and human cancers. *NADSYN1* gene mutations, deletions, amplifications, and multiple alterations in human cancer as demonstrated by the cBioPortal for Cancer Genomics (http://www.cbioportal.org/ accessed on 26 January 2024). The search engine was adjusted to show studies from the TCGA PanCancer Atlas with at least 3% of *NADSYN1* alteration frequency. The “+” sign shows that samples were profiled for structural variants, mutations, and copy number alterations (CNA).

**Table 1 ijms-25-02092-t001:** Summary of the reported NAPRT inhibitors so far and their characteristics.

Name and Structure	Ki/IC50	Mechanism of Action	Anticancer Activity Due to NAPRT Inhibition	Drug-Like Properties
2-Hydroxynicotinic acid 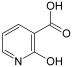	Apparent Ki = 230 µM [84]	Competitive inhibition [84]	Sensitizes NAPRT-expressing ovarian cancer cell lines (OVCAR-5 and OVCAR-8), the pancreatic cancer cell line Capan-1, and the colorectal cell line HCT116 to FK866 [70,89]Cooperates with GNE-617 and GMX1778 in suppressing the tumor growth of head and neck squamous cell carcinoma (RPMI-2650 and Detroit-562) in xenograft-bearing mice [88]2-HNA does not reduce cancer cell viability as a single agent [70,89]	Favorable ADME profile as predicted through computational tools [89]Poor aqueous solubility, which was improved by the generation of its sodium salt 2-HNANa [70]
Ki = 215 ± 5 µM [91]	Competitive inhibition [91]
2-Fluoronicotinic acid 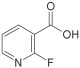	Apparent Ki = 280 µM [84]	Competitive inhibition [84]	Fails to sensitize OVCAR-5 cells to FK866 [91]	N/A
Ki = 149 ± 14 µM [91]	Non-competitive inhibiton [91]
Pyrazinoic acid(Pyrazine-2-carboxylic acid) 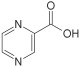	Apparent Ki = 75 µM [84]	Competitive inhibition [84]	Sensitizes OVCAR-5 cells to FK866 by reducing intracellular NAD^+^ contents; the sensitization effect is less pronounced than that of 2-HNA [91]	N/A
Ki = 166 ± 0.14 µM [91]	Non-competitive inhibiton [91]
Salicylic acid 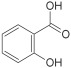	Apparent Ki = 160 µM [85]	Competitive inhibition [85]	Reduces the viability of OVCAR-5 cells without FK866, which is indicative of off-target toxicity [91]	It is a non-steroidal anti-inflammatory drug
Ki = 169 ± 15 µM [91]	Non-competitive inhibiton [91]
Mefenamic acid 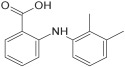	Apparent Ki = 76 µM [85]	Competitive inhibition [85]	N/A	It is a non-steroidal anti-inflammatory drug
Flufenamic acid 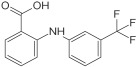	Apparent Ki = 46 µM [85]	Competitive inhibition [85]	N/A	It is a non-steroidal anti-inflammatory drug
Phenylbutazone 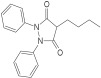	Apparent Ki = 160 µM [85]	Competitive inhibition [85]	N/A	It is a non-steroidal anti-inflammatory drug
4-Hydroxynicotinic acid(Compound **8**) 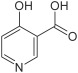	Ki = 307.5µM [89]	Competitive inhibition [89]	Sensitizes NAPRT-expressing ovarian cancer cell lines (OVCAR-5 and OVCAR-8) and the colorectal cell line HCT116 to FK866; it shows a comparable anticancer activity to that of 2-HNA and a better chemosensitizing activity than that of compound **19** [89]	Favorable ADME profile similar to that of 2-HNA but with better water solubility, as predicted through computational tools [89]
1,2,5-thiadiazole-3-carboxylate(Compound **19**) 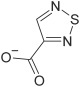	Ki = 295.1µM [89]	Non-competitive inhibition [89]	Sensitizes NAPRT-expressing ovarian cancer cell lines (OVCAR-5 and OVCAR-8) and the colorectal cell line HCT116 to FK866 [89]	Favorable ADME profile as predicted through computational tools; its bioavailability is predicted to be less than that of 2-HNA and of 4-HNA [89]
IM 29 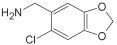	IC50 = 160µM [90]	N/A	Cooperates with FK866 in reducing NAD^+^ levels in OVCAR-5 cells [90]	N/A
CoA 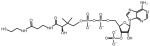	IC50 is about 850 µM [86]	N/A	N/A	CoA is an endogenous co-factor and metabolite
Compound **18** N-(3-(((1,2-dimethyl-1H-benzo [d]imidazol-5-yl)amino)methyl)phenyl)acetamide 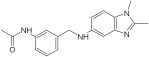	Ki = 338 ± 25 µM toward NA and 134 ± 13 µM toward PRPP [92]	Non-competitive inhibition toward NA and mixed inhibition toward PRPP [92]	N/A	Favorable in vitro ADME features in terms of high kinetic solubility, good metabolic stability, low protein binding, and medium apparent permeability; it is a P-gp substrate [92]

N/A; Not available.

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
