# Peer review of "Inhibitors of NAD+ Production in Cancer Treatment: State of the Art and Perspectives"

_ijms, 2024, doi:10.3390/ijms25042092_

Round 1

Reviewer 1 Report

Comments and Suggestions for Authors

The manuscript “Inhibitors of NAD production in Cancer Treatment: State of the Art and Perspectives” is a review article regarding the homeostasis of nicotinamide adenine dinucleotide and the possibility to modulate the NAD+ synthesis as a therapeutic strategy in cancer treatment. The manuscript may be of interest for the readers.

However, there are important flaws. Addressing the following concerns is mandatory in order to consider the manuscript suitable for publication:

1.       The abbreviation “NAD” must be written as “NAD+” with + in uppercase.

2.       The paragraph of NAD biosynthesis lacks of important information. It should be underlined that it is also possible an extracellular conversion of nicotinamide mononucleotide to nicotinamide riboside by CD73 which represent an important mechanism for maintaining the intracellular NAD+ content (PMID: 32389638).

3.       A paragraph regarding the role of nicotinamide N-methyltransferase (NNMT) in NAD+ homeostasis is mandatory. Indeed, by methylating nicotinamide, NNMT can regulate the NAD+ levels reducing the amount of free nicotinamide which could be converted into NAD+ through the NAD-salvage pathway (PMID: 36829935). NNMT has been reported to be upregulated in many tumors, where it contributes to the tumorigenicity and aggressiveness, and since NNMT can affect NAD+ homeostasis, NAD-dependent enzymes and concentration of SAM, it has a great impact on epigenetics. Notably, a number of NNMT inhibitors are already available and were proposed as a promising strategy for cancer treatment and overcoming chemoresistance (PMID: 34572571; PMID: 34704059; PMID: 34424711).  How these inhibitors would impact on NAD+ levels? The complex interaction of these elements must be taken into account since it has a great impact on NAD+ homeostasis. It is mandatory a discussion.

4.       Authors use abbreviations without specifying their meaning at first use. For instance for NAM (line 77), or NAAD. In figure 1 authors mention 1-meNAM without specifying its meaning. Please revise ALL the manuscript and provide the description of proper abbreviation at first use.

5.       There are terrible mistakes of the chemical structures in figure 1 (e.g. nicotinic acid). Please CORRECT.

6.       The tables are detailed but somehow chaotic. They should be shorten providing only main information, otherwise the reader will be confused.

7.       The quality of figure 3 is poor and should be improved.

I was in doubt whether reject the manuscript or asking for major revisions, due to the critical flaws in chemical structures of molecules, but ultimately decided for the latter.

Comments on the Quality of English Language

Moderate editing of English language required

Reviewer 2 Report

Comments and Suggestions for Authors

Manuscript is well written.

Inhibitors of NAD production in Cancer Treatment: State of the Art and Perspectives

NAD is a crucial coenzyme involved in various cellular processes, including energy metabolism and DNA repair. Disrupting NAD metabolism can impact the survival and growth of cancer cells. Thus, NAD inhibitors have gained attention in the field of cancer research and treatment. The presented review highlights the need of NAD inhibitors and their role in cancer treatment very clearly. The review provides proper background information and clear problem statement. Tables are extremely informative. Usage of references is good. Overall, the manuscript is well structured and well organized.

·         Inclusion of few figures from other research articles would be good.

·         Can also highlight the limitations of real-time or clinical implications of NAD inhibitors.

Round 2

Reviewer 1 Report

Comments and Suggestions for Authors

The manuscript has been improved addressing all the concerns, therefore it can be published.

Comments on the Quality of English Language

English is fine/Minor editing of English language required